# Hybrid AI Intrusion Detection: Balancing Accuracy and Efficiency

**DOI:** 10.3390/s25247564

**Published:** 2025-12-12

**Authors:** Vandit R Joshi, Kwame Assa-Agyei, Tawfik Al-Hadhrami, Sultan Noman Qasem

**Affiliations:** 1Department of Computer Science, Nottingham Trent University, Nottingham NG1 4FQ, UK; vandit.joshi2024@my.ntu.ac.uk (V.R.J.); kwame.assa-agyei@ntu.ac.uk (K.A.-A.); tawfik.al-hdhrami@ntu.ac.uk (T.A.-H.); 2Computer Science Department, College of Computer and Information Sciences, Imam Mohammad Ibn Saud Islamic University (IMSIU), Riyadh 11432, Saudi Arabia

**Keywords:** IoT security, intrusion detection systems, convolutional neural network (CNN), Bi-LSTM, hybrid models, NSL-KDD, UNSW-NB15, performance metrics

## Abstract

The Internet of Things (IoT) has transformed industries, healthcare, and smart environments, but introduces severe security threats due to resource constraints, weak protocols, and heterogeneous infrastructures. Traditional Intrusion Detection Systems (IDS) fail to address critical challenges including scalability across billions of devices, interoperability among diverse protocols, real-time responsiveness under strict latency, data privacy in distributed edge networks, and high false positives in imbalanced traffic. This study provides a systematic comparative evaluation of three representative AI models, CNN-BiLSTM, Random Forest, and XGBoost for IoT intrusion detection on the NSL-KDD and UNSW-NB15 datasets. The analysis quantifies the achievable detection performance and inference latency of each approach, revealing a clear accuracy–latency trade-off that can guide practical model selection: CNN-BiLSTM offers the highest detection capability (F1 up to 0.986) at the cost of higher computational overhead, whereas XGBoost and Random Forest deliver competitive accuracy with significantly lower inference latency (sub-millisecond on conventional hardware). These empirical insights support informed deployment decisions in heterogeneous IoT environments where accuracy-critical gateways and latency-critical sensors coexist.

## 1. Introduction

The rapid integration of Internet of Things (IoT) devices across multiple domains—from smart homes and healthcare monitoring to industrial automation and large-scale sensor networks has fundamentally reshaped modern digital infrastructures.

Despite these advances, IoT ecosystems face severe security vulnerabilities. Most devices operate on constrained hardware, lack built-in security, and are deployed with minimal oversight, making them prime targets for DoS attacks, data leaks, man-in-the-middle exploits, and unauthorized control [1].

Traditional Intrusion Detection Systems (IDS), reliant on centralized processing and high computational overhead, are ill-suited for such environments. They fail to overcome five core challenges: (1) scalability to support millions of heterogeneous devices, (2) interoperability across incompatible protocols (e.g., MQTT, CoAP, Zigbee), (3) real-time detection under microsecond latency constraints, (4) preservation of data privacy in decentralized edge settings, and (5) reliable performance amid highly imbalanced attack distributions.

Artificial Intelligence (AI)-driven IDS leveraging machine learning and deep learning offer adaptive, scalable alternatives capable of learning evolving attack patterns and responding in real time. However, existing solutions remain hindered by high resource demands, inconsistent accuracy, and limited generalizability across diverse IoT contexts (Akif et al., 2025) [2]. This study addresses these gaps through a performance-centric evaluation of lightweight AI models tailored for IoT constraints. The key contributions are as follows: (1) a reproducible and well-performing CNN-BiLSTM baseline that combines 1D-CNN and Bi-LSTM layers; (2) a detailed measurement of inference latency for all three models on the same hardware platform; and (3) an empirical accuracy–latency trade-off analysis to guide model selection in IoT IDS deployments.

This study proposes the development and evaluation of lightweight AI-powered IDS models tailored specifically for IoT environments. The aim is to enhance the detection and classification of malicious activity without compromising device efficiency or network responsiveness. By exploring hybrid modeling techniques and evaluating model effectiveness through metrics like accuracy, precision, recall, F1-score, and latency per sample, the aim of this study is to develop a scalable and efficient framework capable of detecting intrusions within IoT environments. The selection of CNN-BiLSTM, Random Forest, and XGBoost is strategically motivated by their complementary strengths in addressing IoT-specific constraints. CNN-BiLSTM integrates convolutional layers for local spatial feature extraction with bidirectional LSTM for long-range temporal dependency modeling, making it ideal for detecting sophisticated, sequential attack patterns (e.g., multi-stage exploits) prevalent in modern IoT traffic [3,4]. Random Forest, a tree-based ensemble, offers inherent robustness to noisy and imbalanced data common in real-world IoT logs while providing interpretable feature importance rankings without requiring extensive hyper parameter tuning. XGBoost, an optimized gradient boosting framework, delivers state-of-the-art performance with minimal inference latency (sub-millisecond), rendering it suitable for edge deployment on battery-powered sensors. This tri-model evaluation enables a comprehensive performance-centric analysis, quantifying trade-offs between detection capability and computational cost across accuracy-prioritized and latency-critical IoT scenarios.

The structure of this paper is organized as follows: Section 2 discusses previous studies; Section 3 explains the research methodology, covering the experiment setup, NSL-KDD and UNSW-NB15 dataset preparation, the CNN-BiLSTM framework, and Exploratory Data Analysis (EDA); Section 4 outlines the obtained results and their discussion, with evaluations of CNN-BiLSTM, Random Forest, and XGBoost models; Section 5 provides the merged conclusions and future work, summarizing key findings, highlighting model trade-offs for IoT Intrusion Detection Systems (IDS), and outlining potential directions for further investigation.

## 2. Related Work

The growing adoption of Internet of Things (IoT) technologies across critical sectors has increased the demand for resilient, scalable, and interpretable Intrusion Detection Systems (IDS). Recent literature reflects a convergence of Artificial Intelligence (AI), Explainable AI (XAI), deep learning, Federated Learning-based solutions, and optimization techniques to tackle the specific challenges found in IoT systems, including evasion attacks and data security enhancements.

### 2.1. AI and XAI-Driven IDS Models

Explainable AI has become recognized as pivotal theme in IDS research, aiming enhance transparency and user trust. Alabbadi and Bajaber (2025) [5] introduced an XAI-based IDS framework for IoT data streams, demonstrating improved interpretability but lacking scalability analysis. Similarly, Patil et al., 2022 [6] and Arreche et al., 2024 [7] integrated XAI tools such as SHAP and LIME into ensemble models, enabling interpretable alerts for analysts. However, these studies often overlook the trade-offs between explainability and performance, particularly in resource-constrained deployments. Baral et al. (2024) [8] extended this paradigm by incorporating large language models (LLMs), yet complexity of implementation continues to hinder practical deployment.

### 2.2. Deep Learning and Hybrid Architectures

Hybrid deep learning models, especially those combining Convolutional Neural Networks (CNNs) and Long Short-Term Memory (LSTM) networks, have shown promise in enhancing detection accuracy. Altunay and Albayrak (2023) [3] and Gueriani et al., 2024 [4] reported high classification performance using CNN-LSTM architectures on benchmark datasets such as UNSW-NB15 and NSL-KDD. Hossain (2025) [9] further validated the scalability of deep learning models across multiple architectures, including 1D CNNs and RNNs. Liang et al., 2025 [10] introduced a CO-GNN (Combinatorial Optimization Graph Neural Network) framework for heterogeneous secure transmissions in IRS-assisted NOMA communications, optimizing beamforming, power allocation, and phase shifts to maximize secrecy rates against eavesdroppers in dynamic IoT networks; this highlights GNNs’ potential for proactive, scalable security in heterogeneous environments, aligning with the adaptive hybrid models explored in the study. Despite these advances, most studies lack deployment-focused evaluations, leaving questions about latency, memory footprint, and adaptability in heterogeneous IoT environments.

### 2.3. Lightweight and Modular IDS Designs

Efforts to design lightweight IDS models suitable for low-resource devices have gained traction. Jouhari and Guizani (2024) [11] proposed a CNN-BiLSTM model optimized for constrained environments, achieving notable accuracy but struggling with complex attack patterns. Sowmya and Anita (2023) [12] and Faisal and Mubeen (2024) [13] emphasized the importance of adaptive and modular IDS frameworks, though their work remained largely conceptual without empirical validation. Complementing these, Abu Laila (2025) [14] developed a responsive machine-learning framework for preventing evasion attacks in IoT-based IDS, incorporating adversarial training, feature-space transformations, and ensemble detection. This lightweight approach achieved 94.7% accuracy on clean traffic and 89.3% against evasions, with only 15% additional computational overhead, making it ideal for resource-constrained IoT deployments.

### 2.4. Federated Learning and Privacy-Preserving IDS

Researchers have investigated Federated Learning (FL) as a collaborative and privacy-aware approach for intrusion detection. Campos et al., 2022 [15], Javeed et al., 2024 [16], and Ragab et al., 2025 [17] demonstrated the potential of FL-enabled IDS models to maintain data privacy while achieving high detection rates. However, communication overhead, latency, and model convergence issues persist. Liu et al., 2021 [18] and Rampone et al., 2025 [19] attempted to mitigate these challenges by integrating blockchain and near-real-time architectures, yet the complexity of these systems limits their scalability.

### 2.5. Feature Selection and Optimization Techniques

Feature selection remains critical for improving IDS efficiency. Alkahla et al., 2024 [20] and Fang et al., 2023 [21] reviewed filter-based, wrapper-based, and genetic algorithm-driven techniques, showing that optimized feature sets enhance classification accuracy. Bibi et al., 2024 [22] explored pruning and quantization in NLP models, offering insights into model compression strategies, though their applicability to IDS remains underexplored. Alshuaibi et al., 2025 [23] proposed a hybrid genetic algorithm (GA) and hidden Markov model (HMM)-based hashing technique within a block hashing framework to bolster data security against cyber threats like dictionary and brute-force attacks. Drawing from evolutionary biology, the GA incorporates mutation, crossover, and selection operators, paired with Hill cipher encryption using a singular key matrix to minimize hash collisions and reversals. Experimental results highlight its superior resistance to attacks and robustness over existing methods, evaluated via break attempts, hash key size, and algorithmic strength offering a cryptographic layer that complements IDS by ensuring data integrity in IoT preprocessing pipelines. Similarly, Alshinwan et al., 2025 [24] introduced an unsupervised text feature selection approach using an improved Prairie Dog optimization (PDO) algorithm for text clustering, which selects informative feature subsets to mitigate sparsity and uninformative traits in high-dimensional documents. Assessed via K-means clustering against literature benchmarks, it reduces computational time while yielding precise clusters; though focused on text data, this has strong potential for IDS applications, such as optimizing log analysis or anomaly detection in network traffic datasets by enhancing feature relevance and efficiency in unsupervised IoT security tasks. Table 1 provides a comprehensive comparative assessment of recent IoT intrusion detection studies, clearly delineating their objectives, principal contributions, and limitations.

## 3. Methodology

A quantitative experimental design was adopted using the NSL-KDD and UNSW-NB15 dataset. Data preprocessing included normalization, label encoding, and feature selection. Three models, CNN, Bi-LSTM, and hybrid CNN-BiLSTM, were developed and trained using TensorFlow. Performance metrics included accuracy, precision, recall, F1-score, and latency. Benchmarking was conducted against Random Forest and XGBoost classifiers. The CNN-BiLSTM architecture leveraged spatial feature extraction via CNN and temporal sequence learning via Bi-LSTM.

### 3.1. Experimental Setup

Experiments were carried out using a high-capacity computing workstation running Windows 11, equipped with an AMD Ryzen 7 5800X 8-core processor (3.80 GHz), 32 GB DDR4 RAM, a 128-bit Nvidia RTX 4060Ti GPU with 16 GB VRAM (2500 MHz core clock), and 1 TB SSD storage. This configuration provided a robust environment for training and evaluating deep learning models efficiently.

### 3.2. Data Preparation

The study uses the NSL-KDD [25] and UNSW-NB15 [26] datasets. The datasets contain labeled records of both normal and malicious network traffic, covering a broad spectrum of attack types. The data undergoes a structured preparation process to ensure quality and relevance for model training.

Initial steps include preprocessing the dataset by eliminating null entries, duplicate entries, and irrelevant features. Numerical attributes are scaled into a normalized range (commonly 0–1) to ensure uniform influence during training. Categorical variables such as protocol type, service, and connection flags are transformed into numerical representations through methods such as one-hot or label encoding.

Feature selection is guided by statistical correlation, domain expertise, and practical relevance to network behaviors. The final dataset is divided into training and testing portions following an 80:20 proportion, enabling robust model evaluation. This processing ensures the data is clean, balanced, and optimized for deep-learning applications in intrusion detection.

### 3.3. Proposed Model

The proposed model as shown in Figure 1 is a hybrid CNN-BiLSTM model designed to tackle the complex nature of IoT intrusion detection, where attacks often unfold dynamically across multiple dimensions.

Model Explanation:(1)Input Layer: Input Shape (*T*, 1) here *T* is the sequence length and 1 represents single Feature channel. One sequence is fed per sample, but this can be extended to multiple channels when incorporating additional feature types.(2)1D convolution Layer:
(i)Config: 96 Filter, each kernel size of 5.(ii)Activation ‘Relu’, ReLU (F(x)=max(0,x)) [27].(iii)Convolution operation [21].(1)$$y_{i}^{\left(k\right)}=ReLU\left(\sum_{j=0}^{m-1}w_{j}^{\left(k\right)}\cdot x_{i+j}+b^{\left(k\right)}\right)$$Here $w^{\left(k\right)}$ and $b^{\left(k\right)}$ are kernel weights and bias for k-th filter, *m* is kernel size and ReLU activation. $y_{i}^{\left(k\right)}$ is the feature map.(iv)Convolution layer (CNN) captures local temporal patterns, i.e., traffic signatures in network data.(3)MaxPooling1D layer: pool size 2. Reduces sequence length by half, retaining only the most outstanding features. It is used for dimensionality reduction and Noise suppression [28].(4)Bidirectional LSTM layer:
(i)48 units per direction, total 96 states.(ii)Dropout 0.5 input connection and recurrent dropout 0.2 for recurrent connection.(iii)LSTM equation [29]The LSTM updates are computed as follows (all gates and states have dimension 48):Notation:$x_{t}\in R^{d}$: input at time t$h_{t-1}$, $h_{t}\in R^{48}$: previous and current hidden state$c_{t-1}$, $c_{t}\in R^{48}$: previous and current cell state$i_{t}$, $f_{t}$, $o_{t}\in R^{48}$: input, forget, output gates${\tilde{c}}_{t}\in R^{48}$: candidate cell state$W_{i}$, $W_{f}$, $W_{o}$, $W_{c}\in R^{48x(d+48)}$: weight matrices (applied to concatenated $\left[h_{t-1},x_{t}\right]$)$b_{i}$, $b_{f}$, $b_{o}$, $b_{c}\in R^{48}$: bias vectorsσ: sigmoid function⊙: element-wise multiplication[*h*_t−1_, *x*_t_]: concatenation of previous hidden state and current input(2)$$i_{t}=\sigma (W_{i}\cdot \left[h_{t-1},x_{t}\right]+b_{i})\;//\mathrm{Input}\;\mathrm{gate}$$
(3)$$f_{t}=\sigma (W_{f}\cdot \left[h_{t-1},x_{t}\right]+b_{f})\;//\mathrm{forget}\;\mathrm{gate}$$(4)$$o_{t}=\sigma (W_{o}\cdot \left[h_{t-1},x_{t}\right]+b_{o})\;//\mathrm{output}\;\mathrm{gate}$$
(5)$${\tilde{c}}_{t}=\tanh(W_{c}\cdot \left[h_{t-1},x_{t}\right]+b_{c})\;//\mathrm{candidate}\;\mathrm{gate}$$
(6)$$c_{t}=f_{t}\;⊙{\;c}_{t-1}+i_{t}⊙{\tilde{c}}_{t}\;//\mathrm{cell}\;\mathrm{state}\;\mathrm{update}$$
(7)$$h_{t}=o_{t}⊙\tanh(c_{t})\;//\mathrm{Hidden}\;\mathrm{state}\;\mathrm{output}$$(iv)Bidirectional extension [30]:Forward LSTM processes the sequence as follows:(8)$${\vec{h}}_{t}=LSTM(x_{t,}{\vec{h}}_{t-1})$$Backward LSTM processes in reverse order:(9)$${\overset{\leftarrow }{h}}_{t}=LSTM(x_{t,}{\overset{\leftarrow }{h}}_{t+1})$$The final representation is the concatenation:(10)$$h_{t}=[{\vec{h}}_{t};{\overset{\leftarrow }{h}}_{t}]$$This ensure that temporal pattern such as attack bursts followed by silence or l gitimate activity interrupted by anomalies are both captured.(5)Dense layer (64 units, ReLU activation): This layer projects the sequential representations generated by the BiLSTM into a higher-dimensional, non-linear feature space. The ReLU activation introduces sparsity and non-linearity, enabling the network to capture complex interactions across features, such as the joint influence of TCP flags, packet timing, and connection duration, thus yielding richer higher-order representations.(6)Dropout layer: 0.5 dropout rate regularizes the network by randomly disabling neurons during training, this will prevent over fitting [31].(7)Output layer: number of units = number of classes (5 classes in this case), activation SoftMax. SoftMax Equation:

(11)$$P\left(y=c\;|\;x\right)=\frac{\exp\left(z_{c}\right)}{\sum_{j=1}^{C}\exp\left(z_{j}\right)}$$
where $z_{c}$ is the logit for class *c* and *C* is quantity of target classes.

SoftMax ensures outputs are interpretable as class probabilities, facilitating multiclass classification.

The early stop function added to stop the model training when there is no significant improvement on the Val_loss. This will prevent the model from completing the training which can make it overly fit for the dataset.

### 3.4. Hyper Parameter Tuning

Hyper parameter optimization for the CNN-BiLSTM intrusion detection model was conducted using a grid search combined with 5-fold cross-validation on the training subsets of the NSL-KDD and UNSW-NB15 datasets. This process systematically evaluated combinations of key hyper parameters to maximize the validation F1-score, balancing accuracy, generalization, and over-fitting prevention. Early stopping with a patience of 10 epochs was applied, monitoring validation loss to halt training when no improvement was observed. The tuned hyper parameters included one CNN layer to minimize latency without sacrificing accuracy, 96 filters in the CNN layer for optimal temporal feature extraction, 48 LSTM units per direction to balance sequence modeling and efficiency, an Adam learning rate of 0.001 for stable convergence, a batch size of 32 to enhance gradient stability on imbalanced data, and dropout rates of 0.2 for recurrent layers and 0.5 for input and dense layers, reducing validation loss by approximately 15%. A total of 216 hyper parameter combinations were evaluated, with the optimal configuration achieving validation F1-scores of 0.85 on NSL-KDD (multiclass) and 0.97 on UNSW-NB15, resulting in a robust model suitable for IoT environments with computational constraints.

### 3.5. Data Visualization and EDA

Figure 2 depicts the occurrence distribution of various attack types in the NSL-KDD training dataset. The most prevalent label is “normal,” accounting for roughly 67,000 instances. “Neptune” follows as the second-most frequent attack, with around 41,000 occurrences. Compared to the dominant categories, most other attack types show up far less frequently. For instance, “warezclient” appears around 2500 times, while others like “teardrop,” “nmap,” “satan,” “smurf,” and “pod” each fall below 2000. Many additional attacks have fewer than 1000 examples, which clearly points to a strong class imbalance across the dataset.

Figure 2 shows how different attack categories are distributed the UNSW-NB15 dataset. Approximately 56,000 instances are labeled as ‘normal’, followed by around 40,000 occurrences of the ‘generic’ attack type. ‘Exploits’ account for roughly 34,000 instances, while ‘Fuzzer’ and ‘DoS’ attacks register about 18,000 and 13,000, respectively. The ‘Reconnaissance’ category includes close to 10,000 samples. In contrast, all other attack types—such as ‘analysis’, ‘backdoor’, ‘shellcode’, and ‘worms’—each appear fewer than 2000 times, highlighting a significant class imbalance across the dataset.

### 3.6. Feature Importance Analysis

A feature importance analysis was conducted to improve interpretability and understand the decision-making process of the Random Forest and XGBoost models. Using Gini impurity decrease for Random Forest and gain-based importance for XGBoost, the analysis identified the network features most influential in intrusion detection. For the NSL-KDD dataset, both models highlighted traffic volume and connection-based attributes such as src_bytes, dst_bytes, and same_srv_rate as key indicators of probing and DoS attacks, while XGBoost additionally emphasized error-related features like serror_rate and dst_host_srv_serror_rate. For the UNSW-NB15 dataset, both models prioritized flow duration and rate-based features, including sttl, ct_state_ttl, sbytes, and rate, which capture the temporal and throughput characteristics of IoT traffic. XGBoost further weighted stateful and load-based attributes (sload, dload), enhancing its ability to detect complex attack patterns. The models consistently focused on lightweight traffic intensity and flow dynamic features, demonstrating their suitability for efficient, real-time intrusion detection in IoT environments.

Figure 3 presents the distribution of the protocol_type feature within the NSL-KDD and UNSW-NB15 datasets. TCP emerges as dominant protocol, appearing in approximately 18,500 instances in NSL-KDD and around 45,000 instances in UNSW-NB15. UDP ranks second, though significantly less frequent, with roughly 2500 occurrences in NSL-KDD and 30,000 in UNSW-NB15. ICMP is the least represented in NSL-KDD, with just over 1000 examples. Additional protocols such as UNAS, ARP, OSPF, and SCTP are present in the UNSW-NB15 dataset but absent from NSL-KDD. The chart highlights a clear skew in protocol representation, with TCP overwhelmingly dominating the test set. For multiclass classification of attack scenarios, the label-to-class mapping is defined as follows: Class 0 represents normal traffic; Class 1 corresponds to Denial of Service (DoS) attacks; Class 2 captures probing activities; Class 3 denotes Root-to-Local (R2L) intrusions; and Class 4 identifies User-to-Root (U2R) attacks.

### 3.7. Confusion Matrix for Multiclass Classification of Attack Types

Figure 4, Figure 5 and Figure 6 illustrate the model’s classification performance by contrasting predicted outcomes with actual labels. Correct classifications are shown along the diagonal of each confusion matrix, whereas the off-diagonal entries indicate incorrect classification instances, revealing patterns of inter-class confusion. The label-to-class mapping is as follows: Class 0 denotes normal traffic; Class 1 identifies Denial of Service (DoS) attacks; Class 2 captures probing activities; Class 3 corresponds to Root-to-Local (R2L) intrusions; and Class 4 represents User-to-Root (U2R) attacks. CNN-BiLSTM and XGBoost handled the majority classes (Class 0, 1, and 2) effectively, with particular showing very high recall for Class 1. However, the real challenge appeared in the minority classes. CNN-BiLSTM often misclassified Classes 3 and 4 into the larger categories, though it managed to identify a few Class 4 cases and XGBoost showed a similar pattern, where Random Forest failed entirely. Random Forest’s confusion matrix was the most concerning, as it was heavily skewed towards Class 1 and showed almost no recognition of Classes 2, 3, and 4.

### 3.8. Confusion Matrix for Binary Classification:

Both datasets encompass a wide spectrum of attack categories across multiple protocols and services. However, the UNSW-NB15 dataset is more refined in its labeling structure, representing attack traffic as ‘1’ and benign traffic as ‘0’. In contrast, the NSL-KDD dataset lacks this binary classification. To enable consistent training and testing, we manually assigned ‘0’ to normal instances and ‘1’ to all attack types, effectively transforming the labels into a binary format. This process aligns with the principles of one-hot encoding for binary classification tasks.

Figure 7 displays the confusion matrix corresponding to the CNN-BiLSTM model applied across both the NSL-KDD and UNSW-NB15 datasets. In this context, Class 0 represents normal traffic, while Class 1 denotes attack instances. For the NSL-KDD dataset, the model successfully identified 9275 normal instances and 10,889 attack instances, resulting in an overall accuracy near 89% and a precision rate of around 90%. Performance on the UNSW-NB15 dataset was stronger. The model correctly classified all 37,000 normal traffic instances and misclassified only 1122 out of 46,000 attack samples. This corresponds to an accuracy and precision of approximately 98%, demonstrating the resilience and generalization strength of the model across heterogeneous attack scenarios.

Figure 8 shows the confusion matrix for the Random Forest model used in binary classification of attacks. For the NSL-KDD test dataset, the model performed reasonably well, correctly identifying 9342 normal instances and 8456 attack instances. This translates to an accuracy and precision of about 78%. However, its performance on the UNSW-NB15 dataset was noticeably weaker. The confusion matrix indicates that numerous normal samples were incorrectly labeled as attacks, potentially increasing the false alarm rate in practical intrusion detection applications.

Figure 9 presents the confusion matrix for the XGBoost model under binary classification of attack instances. Findings indicate that the model achieves satisfactory performance on Class 0 (normal traffic) in the NSL-KDD dataset, with relatively few misclassifications outperforming both Random Forest and CNN-BiLSTM in this category. However, its performance on Class 1 (attack traffic) is less effective, with the lowest correct prediction rate among the three models. A similar trend is observed with the UNSW-NB15 dataset: while the model excels in identifying normal traffic, it struggles to accurately classify attack instances, indicating limitations in its generalization across diverse intrusion patterns.

## 4. Discussion

The IDS models were evaluated using the following performance metrics:

### 4.1. CNN-BiLSTM Model

#### 4.1.1. Model Accuracy

Figure 10 illustrates a rapid rise in training accuracy (Train_Acc) with multiclass attack NSL-KDD, which remains consistently high across all epochs signifying effective learning from the training data. In contrast, validation accuracy (Val_Acc) starts around 0.82 but fluctuates between 0.76 and 0.81 throughout the training process. This persistent gap between training and validation performance highlights a generalization issue: while the model performs reliably on known data, its ability to handle unseen inputs is less stable. The observed variability in validation accuracy implies that certain attack categories may not be consistently detected across epochs.

Figure 11 shows how the model performed when using binary classification on both the NSL-KDD and UNSW-NB15 datasets. For NSL-KDD, validation accuracy stayed fairly consistent, ranging from 0.83 to 0.89 across epochs. On the other hand, the UNSW-NB15 dataset showed a sharp improvement, with validation accuracy rising from 0.77 to 0.98. The number of epochs shown is limited due to the use of an early stopping function, which automatically halts training when there is no meaningful improvement in validation loss. This helps prevent over fitting and ensures the model does not continue training unnecessarily.

#### 4.1.2. Model Loss

As shown in Figure 12, the training loss begins at approximately 0.15 and quickly drops to near zero, indicating minimal error and strong convergence during multiclass classification on the NSL-KDD training set. In contrast, the validation loss starts around 1.0 and steadily increases, peaking near 5.0 with noticeable fluctuations. This widening gap between training and validation losses suggests noticeable overfitting, where the model becomes tailored to training data yet fails to generalize well to new samples.

Figure 13 shows how the model performed during binary classification training using benchmark datasets; the NSL-KDD and UNSW-NB15 datasets. The CNN framework showed steady improvement in validation loss. For NSL-KDD, the Val_loss stayed between 0.5 and 0.9, which suggests the model was learning effectively without signs of over fitting. On the UNSW-NB15 dataset, Val_loss started at 1.1, briefly spiked to 1.3, and then dropped sharply to around 0.2–0.1. This drop indicates that the model successfully captured training patterns and generalized effectively to previously unseen data.

### 4.2. Random Forest Model

The Random Forest model was evaluated on both the NSL-KDD and UNSW-NB15 datasets for multiclass and binary attack detection tasks. As shown in Table 2, the model achieved an accuracy of 63.74% for multiclass classification on the NSL-KDD dataset, with corresponding recall and F1-score values of 0.6374 and 0.5474, respectively. The relatively low precision (0.4994) indicates a tendency to misclassify some attack categories, reflecting moderate discriminative capability across multiple classes.

For binary classification, performance improved significantly. On the NSL-KDD dataset, the Random Forest achieved an accuracy of 78.95%, precision of 83.88%, and an F1-score of 0.7880, suggesting a balanced and reliable detection of attack versus normal traffic. Similarly, on the UNSW-NB15 dataset, accuracy reached 75.53% with precision of 83.06% and F1-score of 0.7318. Although slightly lower than on NSL-KDD, this still demonstrates reasonable generalization. The minor performance drop may stem from dataset heterogeneity and variations in attack distribution.

### 4.3. XGBoost Model

As summarized in Table 2, the XGBoost model also exhibited strong classification performance across both datasets. For multiclass attack detection on the NSL-KDD dataset, it achieved an accuracy of 77.69%, precision of 79.19%, recall of 77.69%, and F1-score of 0.7287. These results indicate that XGBoost effectively balances precision and recall in identifying diverse attack types.

In binary classification tasks, XGBoost demonstrated robust generalization. On the NSL-KDD dataset, it reached 77.30% accuracy, 83.51% precision, and an F1-score of 0.7701. Notably, when applied to the UNSW-NB15 dataset, the model’s performance improved substantially, with accuracy and recall of 96.89%, precision of 97.09%, and F1-score of 0.9690. These outcomes show that XGBoost performs competitively with deep learning architectures, particularly on more balanced datasets like UNSW-NB15.

### 4.4. Latency per Sample

The latency per sample was calculated by dividing the total inference time by the number of test instances, providing a standardized measure of computational efficiency. As reflected in Table 2, traditional machine-learning models such as Random Forest and XGBoost demonstrated extremely low latency (approximately 0.0002–0.0008 ms/sample), making them suitable for real-time intrusion detection. In contrast, the CNN-BiLSTM models exhibited slightly higher latency (ranging from 0.059 to 0.126 ms/sample), which is expected given their deeper architecture and sequential processing nature. Despite this, all models maintained acceptable inference speeds suitable for deployment in IoT-based IDS environments.

### 4.5. Discussion of Results

Table 2 presents a detailed comparison of three models CNN-BiLSTM, Random Forest, and XGBoost evaluated across multiclass and binary attack classification tasks employing both the NSL-KDD and UNSW-NB15 benchmark datasets. The analysis considers core performance metrics (F1-score, recall, precision, accuracy) alongside latency per sample, offering a holistic view of each model’s predictive capability and computational efficiency. The CNN-BiLSTM model consistently delivered top-tier classification performance across all tasks. On the UNSW-NB15 dataset, it achieved near-perfect scores (F1: 0.9864, Accuracy: 0.9864), demonstrating its strength in capturing complex attack patterns. Binary classification on NSL-KDD also showed strong results (F1: 0.8949), while multiclass performance remained competitive (F1: 0.7680). However, this accuracy came with a trade-off in latency. Inference times ranged from 0.0590 ms to 0.1261 ms per sample, the highest among all models. The deeper architecture and sequential processing of CNN-BiLSTM contribute to this overhead, rendering it less practical for real-time use in environments where latency is critical.

Random Forest showed mixed performance. It performed reasonably well in binary classification on NSL-KDD (F1: 0.7880), with strong precision (0.8388), but struggled with multiclass classification (F1: 0.5474), suggesting limitations in handling more granular attack categories.

Its standout feature is speed. With latency as low as 0.0002 ms per sample, Random Forest offers rapid inference, making it attractive for systems where computational resources are constrained or response time is critical. However, its lower predictive performance, especially on multiclass tasks, may limit its utility in high-stakes detection scenarios.

XGBoost emerged as the most balanced model in the study. It delivered strong classification metrics across both datasets and tasks, particularly on UNSW-NB15 (F1: 0.9690, Precision: 0.9709), rivaling CNN-BiLSTM in accuracy while outperforming it in latency.

Its multiclass performance on NSL-KDD (F1: 0.7287) was also robust, and binary classification results were consistent (F1: 0.7701). With latency as low as 0.0001 ms per sample, XGBoost offers exceptional computational efficiency without compromising predictive quality, making it a viable solution for real-time Intrusion Detection Systems.

However, although NSL-KDD and UNSW-NB15 are well-established benchmarks with diverse attack taxonomies enabling reproducible comparisons, neither was specifically designed for IoT scenarios, NSL-KDD reflects legacy enterprise traffic and UNSW-NB15 focuses on general LAN/WAN environments, potentially limiting model generalization to real IoT networks characterized by device heterogeneity, low-power protocols (e.g., MQTT, CoAP), and extreme class imbalance; consequently, the observed minority-class weaknesses may underestimate challenges in actual IoT deployments, where rare protocol-specific attacks (e.g., Mirai botnets) could yield higher false negatives, underscoring the need for future validation on IoT-native datasets despite the current findings positioning CNN-BiLSTM for accuracy-critical contexts and XGBoost for latency-sensitive real-time IDS. Table 3 maps the limitations of previous studies to the key findings and contributions of the present work.

## 5. Conclusions and Future Work

This work systematically compared a hybrid CNN-BiLSTM model with two lightweight machine-learning baselines (Random Forest and XGBoost) for intrusion detection in IoT environments. The CNN-BiLSTM consistently achieved the highest detection performance, with F1-scores of 0.7680, 0.8949, and 0.9864 supported by strong precision and recall. However, its deep architecture introduced higher inference latency (0.0590–0.1261 ms). XGBoost offered the best balance between accuracy and computational efficiency, maintaining competitive F1-scores (up to 0.9690) and the lowest latency (0.0001–0.0002 ms), while Random Forest provided fast inference but more variable performance, especially for multiclass tasks. Confusion matrix analysis showed that all models struggled with minority attack classes, emphasizing the impact of dataset imbalance. Overall, CNN-BiLSTM is well suited to accuracy-critical IoT environments, whereas XGBoost is more appropriate for latency-sensitive deployments. These findings clarify the trade-offs between predictive capability and efficiency when designing scalable IDS solutions for heterogeneous IoT ecosystems.

Future work should focus on improving model interpretability through explainable AI, enabling privacy-preserving training via Federated Learning, and validating IDS models on realistic IoT-specific datasets such as IoT-23, CIC-IoT2023, and TON_IoT. Addressing class imbalance remains a priority; techniques such as SMOTE, class-weighted loss functions, and other imbalance-aware training strategies could enhance detection of rare and zero-day attacks. Additional optimizations including pruning, quantization, and edge-oriented deployment strategies—may further improve scalability and efficiency in real-world IoT scenarios.

## Figures and Tables

**Figure 1 sensors-25-07564-f001:**
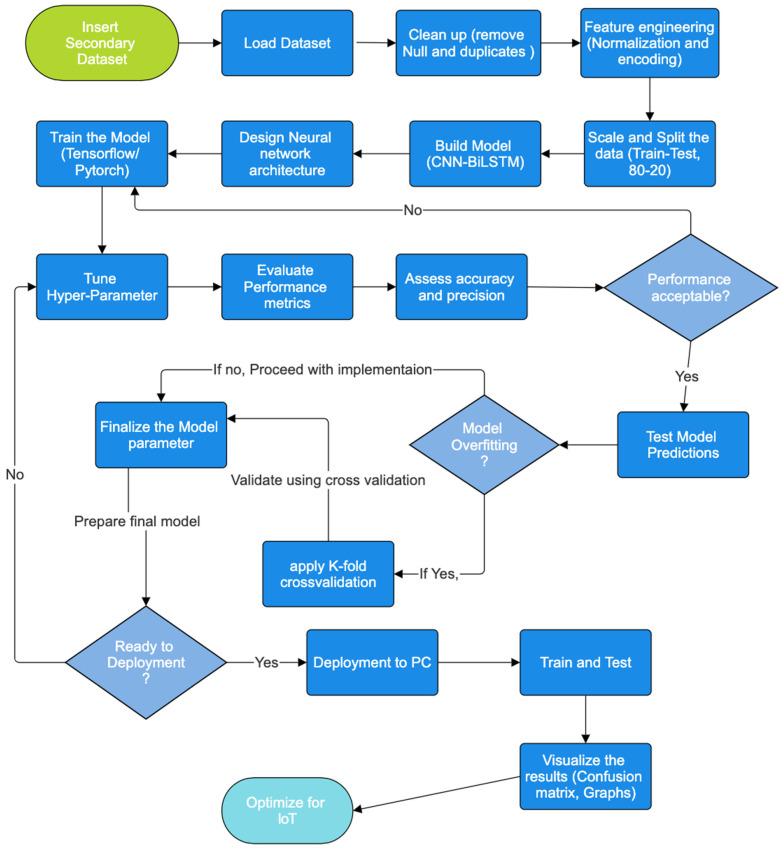
Flow Chart of CNN-BiLSTM.

**Figure 2 sensors-25-07564-f002:**
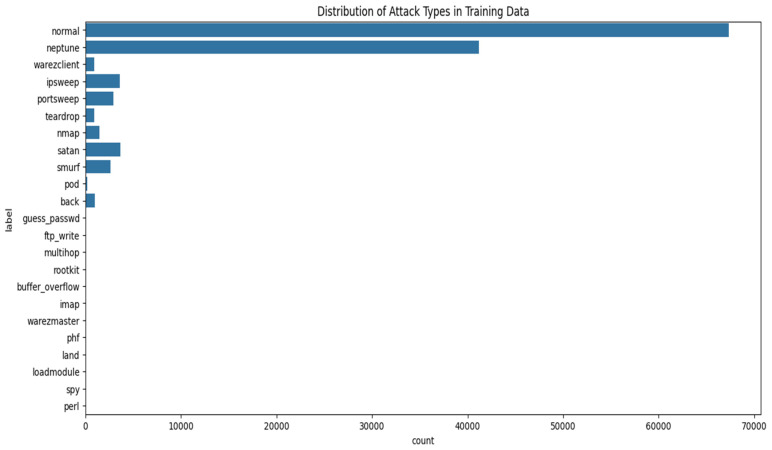
Distribution of attack types in training dataset.

**Figure 3 sensors-25-07564-f003:**
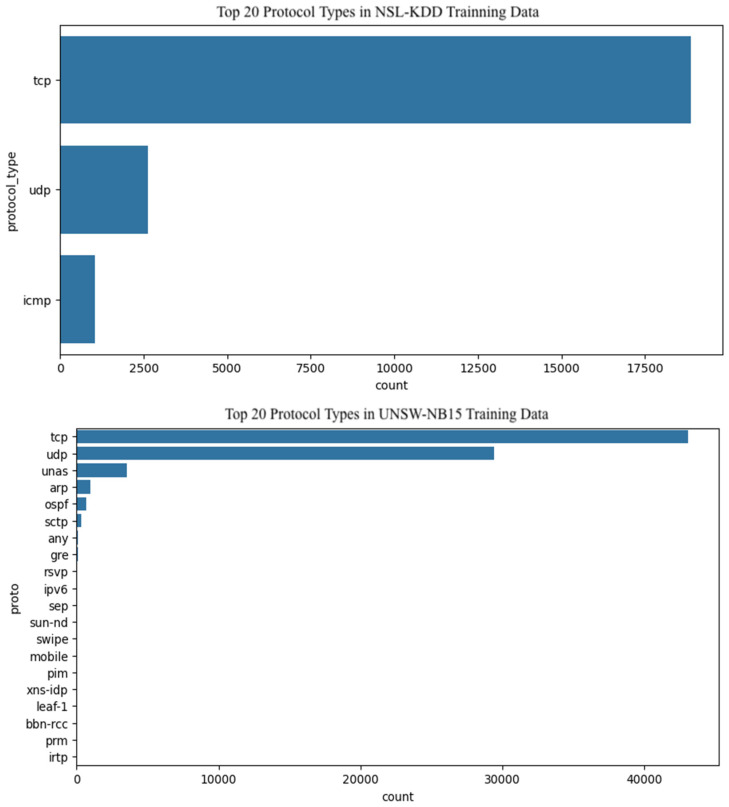
Top 20 Protocol Types.

**Figure 4 sensors-25-07564-f004:**
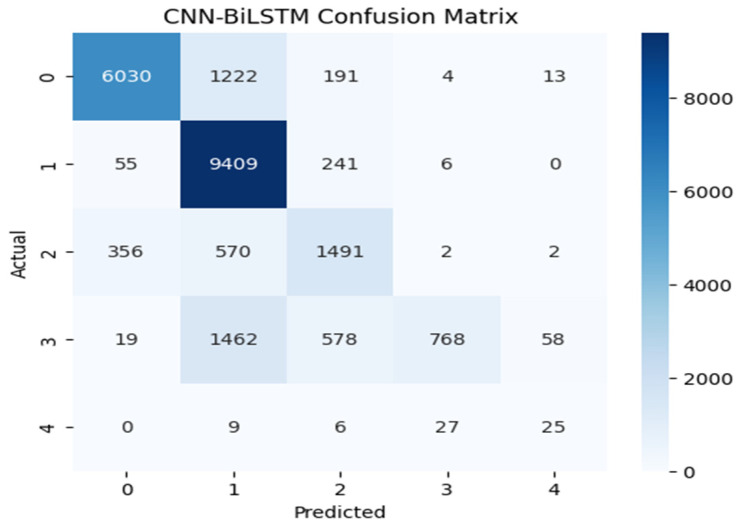
CNN-BiLSTM confusion matrix.

**Figure 5 sensors-25-07564-f005:**
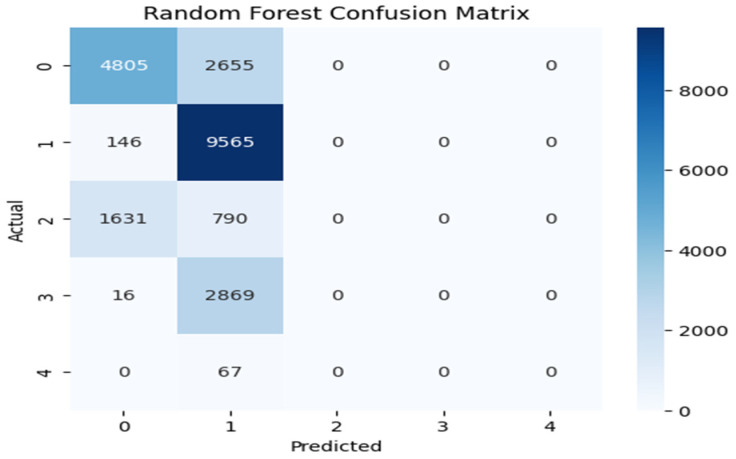
Random Forest confusion matrix.

**Figure 6 sensors-25-07564-f006:**
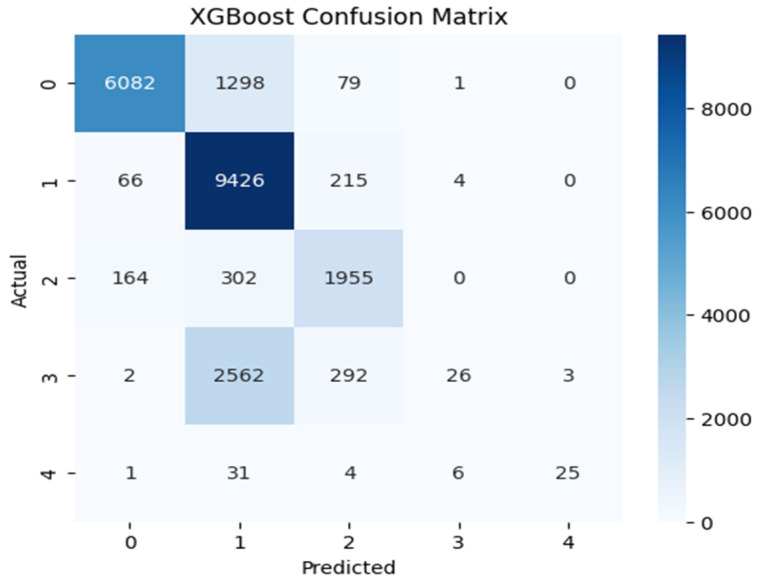
XGBoost model confusion matrix.

**Figure 7 sensors-25-07564-f007:**
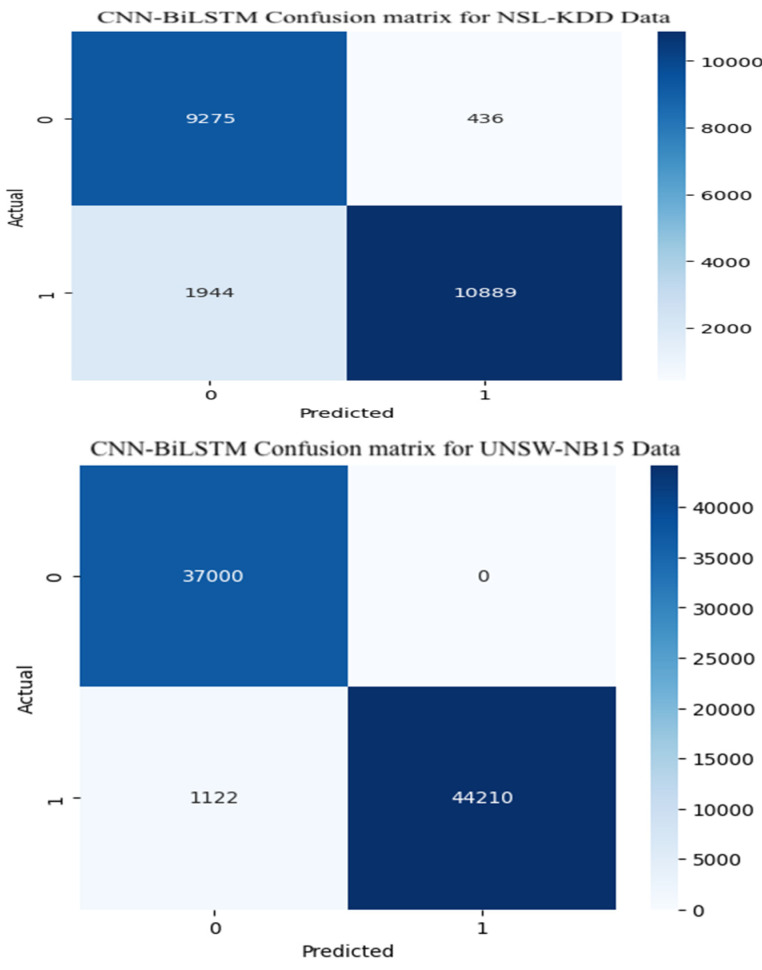
Confusion matrix for CNN-BiLSTM.

**Figure 8 sensors-25-07564-f008:**
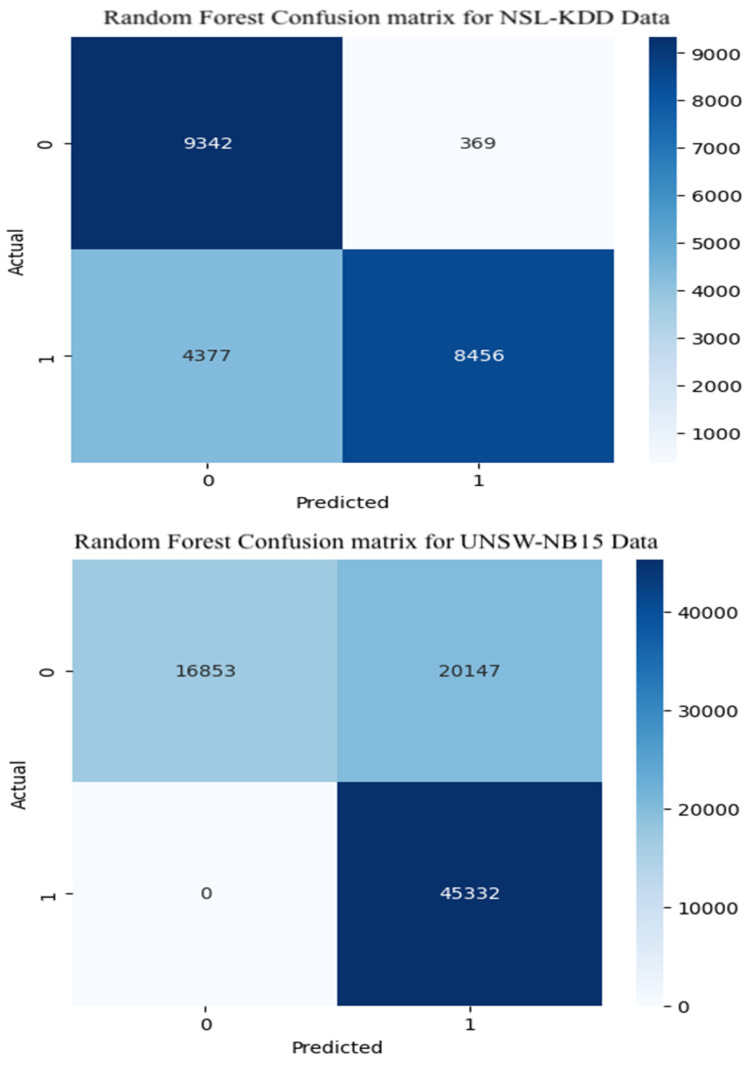
Random Forest confusion matrix.

**Figure 9 sensors-25-07564-f009:**
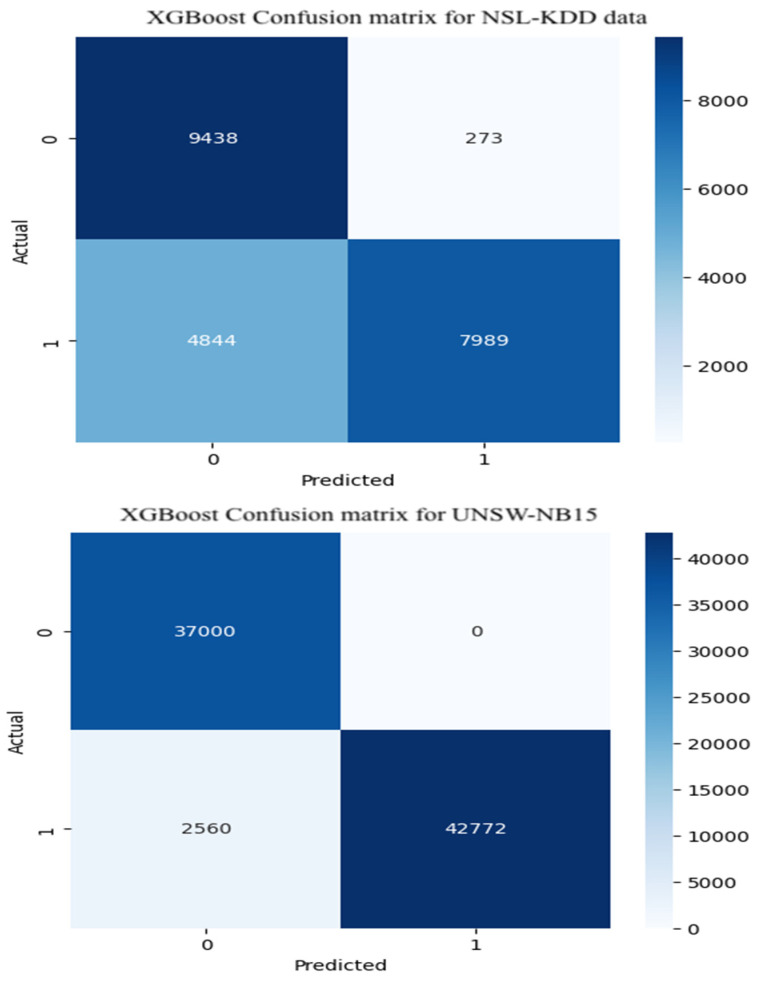
XGBoost confusion matrix.

**Figure 10 sensors-25-07564-f010:**
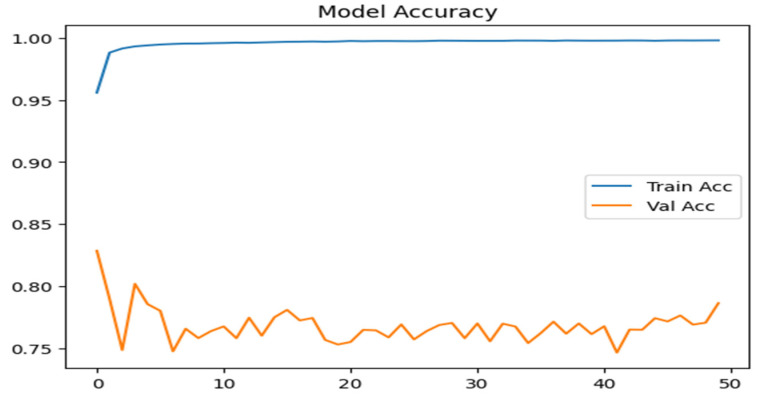
Model accuracy for multiclass attacks.

**Figure 11 sensors-25-07564-f011:**
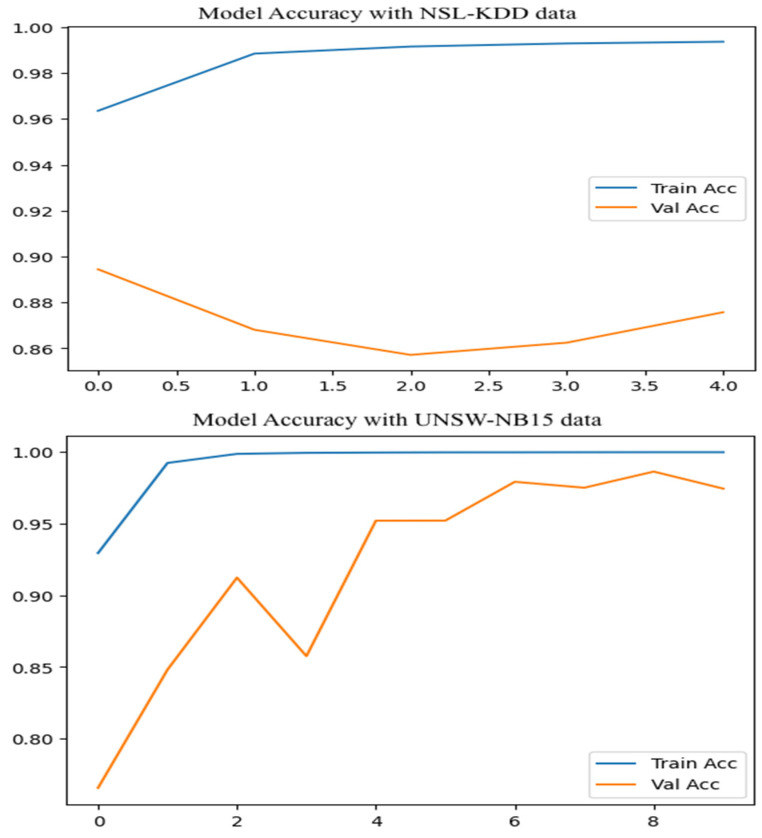
Model accuracy on Binary class attacks.

**Figure 12 sensors-25-07564-f012:**
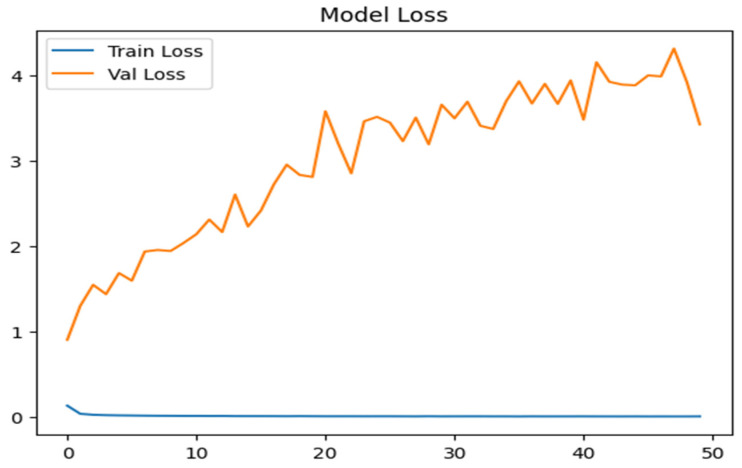
Model loss for multiclass attacks.

**Figure 13 sensors-25-07564-f013:**
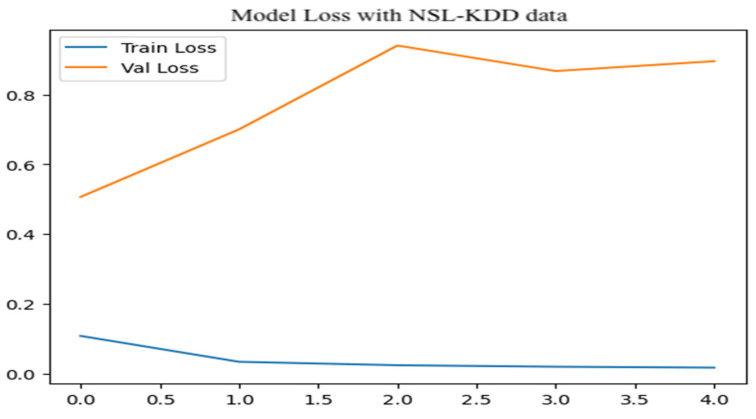
Model loss for binary class attack.

**Table 1 sensors-25-07564-t001:** Comparative Analysis of Related Studies on IoT Intrusion Detection.

Study	Objective	Key Contribution/Gap Addressed	Limitations
Alabbadi & Bajaber (2025) [5]	Develop XAI-enhanced IDS for real-time IoT data streams	Improved interpretability using SHAP/LIME on streaming data	No scalability testing; not evaluated on edge devices; missing methodological details prevent replication or deeper analysis
Patil et al., 2022 [6]; Arreche et al., 2024 [7]	Integrate XAI into ensemble IDS models	Transparent decision-making via feature attribution	Ignores latency and resource overhead in constrained environments; missing methodological details prevent replication or deeper analysis
Altunay & Albayrak (2023) [3]; Gueriani et al., 2024 [4]	High-accuracy hybrid CNN + LSTM for attack classification	Strong performance on UNSW-NB15/NSL-KDD	No latency analysis; not optimized for edge deployment; missing methodological details prevent replication or deeper analysis
Jouhari & Guizani (2024) [11]	Lightweight CNN-BiLSTM for resource-constrained devices	Reduced model size with acceptable accuracy	Struggles with complex, multi-stage attacks and missing methodological details prevent replication or deeper analysis
Abu Laila (2025) [14]	Evasion-resistant lightweight IDS with adversarial robustness	94.7% accuracy, only 15% overhead versus baseline	Limited to binary classification, no multiclass evaluation, and missing methodological details prevent replication or deeper analysis
Campos et al., 2022 [15]; Javeed et al., 2024 [16]	Privacy-preserving IDS via Federated Learning	High accuracy with no raw data sharing	High communication overhead, convergence instability, and missing methodological details prevent replication or deeper analysis
Rampone et al., 2025 [19]	Near-real-time FL with blockchain integration	Enhanced privacy and auditability	Increased system complexity; scalability concerns
Alkahla et al., 2024 [20]; Fang et al., 2023 [21]	Feature selection for IDS efficiency	Reduced dimensionality, improved accuracy	Not integrated into end-to-end lightweight models, missing methodological details prevent deeper analysis
Alshuaibi et al., 2025 [23]	GA + HMM-based secure hashing for data integrity	Strong resistance to dictionary/brute-force attacks	Complementary (not core IDS); adds preprocessing overhead; missing methodological details prevent deeper analysis
Alshinwan et al., 2025 [24]	Unsupervised feature selection via PDO	Efficient clustering in high-dimensional data	missing methodological details prevent deeper analysis
Liang et al., 2025 [10]	Secure NOMA transmission using CO-GNN	Optimized secrecy rate in IRS-assisted networks	Not an IDS; focuses on physical-layer security

**Table 2 sensors-25-07564-t002:** Experimental Model Benchmark comparison.

Model	Dataset	F1-Score	Recall	Precision	Accuracy	Latency per Sample (ms)
CNN-BiLSTM	NSL-KDD Multiclass attack	0.7680	0.7862	0.8151	0.7862	0.0668
	NSL-KDD Binary class attack	0.8949	0.8944	0.9034	0.8944	0.0590
	UNSW-NB15	0.9864	0.9864	0.9868	0.9864	0.1261
Random Forest	NSL-KDD Multiclass attack	0.5474	0.6374	0.4994	0.6374	0.0008
	NSL-KDD Binary class attack	0.7880	0.7895	0.8388	0.7895	0.0002
	UNSW-NB15	0.7318	0.7553	0.8306	0.7553	0.0002
XGBoost	NSL-KDD Multiclass attack	0.7287	0.7769	0.7919	0.7769	0.0002
	NSL-KDD Binary class attack	0.7701	0.7730	0.8351	0.7730	0.0001
	UNSW-NB15	0.9690	0.9689	0.9709	0.9689	0.0001

**Table 3 sensors-25-07564-t003:** Addressing Limitations of Prior IoT-IDS Research: Key Findings and Contributions of the Present Study.

Limitations of Previous Studies	Key Findings of Our Study	Contributions of Our Study
Lack of latency benchmarking and edge deployment analysis	XGBoost achieves F1 = 0.9690 (UNSW-NB15) with 0.0001 ms/sample latency	Low-latency XGBoost model suitable for real-time edge inference on constrained IoT devices
High computational overhead of deep models; no accuracy–latency trade-off analysis	CNN-BiLSTM delivers F1 = 0.9864 (UNSW-NB15 binary), 0.8949 (NSL-KDD binary) at 0.059–0.126 ms/sample	Fully documented hybrid CNN-BiLSTM architecture with equations, tuning, and performance metrics for complex attack detection
No systematic comparison between deep and lightweight models	All models evaluated across multiclass and binary tasks on NSL-KDD and UNSW-NB15 with confusion matrices, EDA, and feature importance	Tri-model benchmarking framework (CNN-BiLSTM, Random Forest, XGBoost) enabling performance-centric IDS design
Absence of context-aware deployment guidelines for heterogeneous IoT	XGBoost offers best accuracy–speed balance; CNN-BiLSTM excels in accuracy-critical scenarios	Performance-centric deployment strategy: Use CNN-BiLSTM in accuracy-critical gateways, XGBoost in latency-sensitive sensors

## Data Availability

The research data supporting this publication are accessible through the project’s GitHub repository https://github.com/kwameassa/Vandit_IDS_Project (accessed on 31 August 2025).

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
