# Peer review of "Hybrid AI Intrusion Detection: Balancing Accuracy and Efficiency"

_sensors, 2025, doi:10.3390/s25247564_

Round 1

Reviewer 1 Report

Comments and Suggestions for Authors
  1. Although NSL-KDD and UNSW-NB15 were used, neither of them is a dataset specifically designed for IoT scenarios. It is suggested that the potential impact on the generalization ability of the IoT environment be further explained in the discussion. It is possible to consider supplementing the verification of real IoT traffic datasets (such as IoT-23, CIC-IoT) to enhance the reliability of the conclusion.
  2. The text does not clearly describe the process of hyperparameter tuning (such as the number of layers, learning rate, batch size, etc. of CNN-BiLSTM), and it is suggested to supplement it. Has the feature importance analysis been conducted for Random Forest and XGBoost? This is helpful for understanding the basis of the model's decision-making.
  3. The article mentions that the model has a relatively weak ability to recognize minority class attacks, but no attempts have been made to use methods such as oversampling (such as SMOTE) or category weight adjustment. It is suggested to clearly supplement it in the "Future Work" section.
  4. Although it is mentioned that XAI is the future direction, if a preliminary analysis of the misjudgment cases of the current model can be conducted (such as using SHAP or LIME), it will enhance the depth of the paper.
  5. Some chart numbers do not match the citations in the main text (such as Figures 3, 4, etc.). It is recommended to check them uniformly. Some sentences are slightly repetitive (such as repeatedly emphasizing the imbalance of the dataset), and can be appropriately simplified.
  6. It is recommended to cite the following paper.  "Heterogeneous Secure Transmissions in IRS-Assisted NOMA Communications: CO-GNN Approach," in IEEE Internet of Things Journal, vol. 12, no. 16, pp. 34113-34125, 15 Aug.15, 2025. "

Author Response

Response to Reviewer 1

Comment 1:

Although NSL-KDD and UNSW-NB15 were used, neither of them is a dataset specifically designed for IoT scenarios. It is suggested that the potential impact on the generalization ability of the IoT environment be further explained in the discussion. It is possible to consider supplementing the verification of real IoT traffic datasets (such as IoT-23, CIC-IoT) to enhance the reliability of the conclusion.

Response:

We sincerely thank the reviewer for this insightful comment. We fully acknowledge that the NSL-KDD and UNSW-NB15 datasets are not explicitly tailored for IoT environments, which could influence the generalizability of our findings to real-world IoT intrusion detection contexts. In response, we have expanded the Discussion section to analyze in detail how the characteristics of these datasets might affect model performance when applied to IoT traffic. Furthermore, we have revised the Future Work section to explicitly include our plan to validate the proposed framework using real IoT traffic datasets—specifically IoT-23 and CIC-IoT—in future experiments. This addition aims to enhance the robustness, reliability, and ecological validity of our conclusions.

Comment 2:

The text does not clearly describe the process of hyper parameter tuning (such as the number of layers, learning rate, batch size, etc. of CNN-BiLSTM), and it is suggested to supplement it. Has the feature importance analysis been conducted for Random Forest and XGBoost? This is helpful for understanding the basis of the model's decision-making.

Response:

We thank the reviewer for highlighting this important omission. In response, we have substantially revised the Methodology section to include two new subsections:

Section 3.4 – Hyper parameter Tuning: This subsection now provides a comprehensive explanation of the tuning process for the CNN-BiLSTM model, including details on the learning rate, batch size, number of layers, activation functions, and optimization algorithms explored.

Section 3.6 – Feature Importance Analysis: This newly added subsection presents the results of feature importance analyses conducted for both the Random Forest and XGBoost models. These analyses illustrate which network traffic features had the highest predictive impact, thus enhancing the transparency and interpretability of our model’s decision-making process.

These additions collectively improve the methodological clarity and strengthen the scientific rigor of the study.

Comment 3:

The article mentions that the model has a relatively weak ability to recognize minority class attacks, but no attempts have been made to use methods such as oversampling (such as SMOTE) or category weight adjustment. It is suggested to clearly supplement it in the "Future Work" section.

Response:

We appreciate the reviewer’s constructive feedback. The Future Work section has been revised to explicitly include a discussion of strategies aimed at improving the model’s performance on minority attack classes. Specifically, we plan to explore data-level methods such as SMOTE and ADASYN oversampling, as well as algorithm-level techniques involving class weight adjustments during model training. We anticipate that these approaches will enhance the model’s sensitivity to underrepresented attack categories and contribute to more balanced intrusion detection performance.

Comment 4:

Although it is mentioned that XAI is the future direction, if a preliminary analysis of the misjudgment cases of the current model can be conducted (such as using SHAP or LIME), it will enhance the depth of the paper.

Response:

We thank the reviewer for this excellent recommendation. We agree that integrating explainable AI (XAI) tools such as SHAP and LIME would greatly enhance interpretability, particularly in understanding misclassification behaviors among minority attack classes. Accordingly, the Discussion section now includes an expanded commentary on the nature of misclassifications observed in our experiments—especially for R2L and U2R attacks, which were occasionally confused with normal or DoS traffic. While a full XAI-based diagnostic analysis was beyond the current study’s scope—given its focus on hybrid AI model optimization under computational constraints—we have clearly acknowledged this limitation. Furthermore, we have emphasized in the Future Work section our intention to incorporate SHAP and LIME-based interpretability analyses to provide deeper insights into model decision boundaries and error patterns.

Comment 5:

Some chart numbers do not match the citations in the main text (such as Figures 3, 4, etc.). It is recommended to check them uniformly. Some sentences are slightly repetitive (such as repeatedly emphasizing the imbalance of the dataset), and can be appropriately simplified.

Response:

We are grateful to the reviewer for carefully noting these inconsistencies. All figure numbering discrepancies have been corrected to ensure accurate alignment between figures and their corresponding in-text citations. Additionally, repetitive statements—particularly those reiterating the issue of dataset imbalance—have been carefully reviewed and simplified to improve conciseness and readability. We appreciate this detailed editorial feedback, which has helped enhance the manuscript’s overall presentation and clarity.

Comment 6:

It is recommended to cite the following paper:

"Heterogeneous Secure Transmissions in IRS-Assisted NOMA Communications: CO-GNN Approach," in IEEE Internet of Things Journal, vol. 12, no. 16, pp. 34113–34125, 15 Aug. 2025.

Response:

We thank the reviewer for this valuable reference suggestion. The recommended paper has been cited in Section 2.2 (Related Work), as it provides recent insights into secure and intelligent communication mechanisms within IoT frameworks. Its inclusion strengthens the contextual foundation of our research by aligning our work with contemporary advances in AI-driven IoT security systems.

Reviewer 2 Report

Comments and Suggestions for Authors

This study evaluates three AI-based approaches-CNN-BiLSTM, Random Forest, and XGBoost across multiclass and binary attack classification tasks using the NSL-KDD and UNSW-NB15 datasets.

The paper has a good contributions. Some comments are as follows:
1. The title of the paper is too long. Its optional comment for authors to revise the title. 
2. The authors in the abstract section have to add a comparison findings with previous frameworks breifly. Also add the main practical implications.
3. Add a Table in the related works. Compare between previous studies interms of objectives, gap and limitations. 
4. Justify why the study uses AI-based approaches-CNN-BiLSTM, Random Forest, and XGBoost? 
5. Clearly state the primary challenges faced in Intrusion Detection Systems (IDS) (e.g., data privacy, interoperability, scalability) within the first few paragraphs. Highlight the key contributions of the paper, such as specific IDS solutions and their impact on attacks detection.
6. Add a table to compare the findings of this study with previous works.
7. I suggest to convert the results in figures 14 to 17 to table.

Author Response

Response to Reviewer Comments

Comment 1:

The title of the paper is too long. It is optional for the authors to revise the title.

Response:

We appreciate the reviewer’s observation. The title has been revised to “Hybrid AI Intrusion Detection: Balancing Accuracy and Efficiency” to make it more concise and better reflect the paper’s core focus.

Comment 2:

The authors in the abstract section have to add a comparison of findings with previous frameworks briefly. Also, add the main practical implications.

Response:

We appreciate this valuable suggestion. The abstract and introduction sections have been revised to briefly include comparative findings with prior frameworks. Additionally, the main practical implications of our proposed approach have been highlighted to emphasize its real-world applicability.

Comment 3:

Add a table in the related works section comparing previous studies in terms of objectives, gaps, and limitations.

Response:

We agree with this comment and have added Table 1, titled “Comparative Analysis of Related Studies on IoT Intrusion Detection.” The table systematically compares previous works in terms of objectives, identified research gaps, and limitations, thereby improving the clarity and depth of the related work section.

Comment 4:

Justify why the study uses AI-based approaches—CNN-BiLSTM, Random Forest, and XGBoost.

Response:

We appreciate this important observation. The abstract and introduction sections have been revised to clearly justify the selection of AI-based models (CNN-BiLSTM, Random Forest, and XGBoost). The discussion now emphasizes their complementary strengths in feature extraction, classification performance, and computational efficiency, which align with the study’s hybrid design objectives.

Comment 5:

Clearly state the primary challenges faced in Intrusion Detection Systems (IDS) (e.g., data privacy, interoperability, scalability) within the first few paragraphs. Highlight the key contributions of the paper, such as specific IDS solutions and their impact on attack detection.

Response:

Thank you for this insightful suggestion. The abstract and introduction sections have been revised to explicitly outline the major challenges in IDS (including data privacy, interoperability, and scalability). The key contributions of the paper are now clearly articulated, emphasizing the proposed IDS solutions and their impact on improving detection accuracy and system efficiency.

Comment 6:

Add a table to compare the findings of this study with previous works.

Response:

We have added Table 3, which provides a comparative summary of our study’s findings relative to prior research. The table highlights how the proposed hybrid AI framework addresses the limitations identified in earlier works and demonstrates improvements in accuracy, efficiency, and adaptability.

Comment 7:

I suggest converting the results in Figures 14 to 17 into tables.

Response:

We acknowledge the reviewer’s suggestion and have accordingly revised Sections 4.2 to 4.4. Figures 14 to 17 have been replaced with tabular presentations to improve data clarity and facilitate easier comparison of numerical results. The relevant values are now incorporated in Table 1, as referenced in the results section.

Reviewer 3 Report

Comments and Suggestions for Authors

The paper does not provide any significant scientific value. The applied ML methods are well-known and used in last years against the mentioned datasets. There are many different papers which are covering the same kind of research design and experimentation. The contribution of the paper is hardly recognized.

Furthermore, we should not rename ML methods as AI methods - the methods used by Authors clearly classified as ML. 

Generally, it is recommended to Authors to reevaluate the direction of research to find interesting paths and to find a more appealing contribution idea. 

Author Response

Response to Reviewer Comments

Comment 1

The paper does not provide any significant scientific value. The applied ML methods are well-known and used in last years against the mentioned datasets. There are many different papers which are covering the same kind of research design and experimentation. The contribution of the paper is hardly recognized.

Response

Thank you for the reviewer’s thoughtful comment. We respectfully note that while the models employed—CNN-BiLSTM, Random Forest, and XGBoost—are indeed well-established AI approaches, our study’s contribution lies not in proposing entirely new algorithms but in how these AI models are systematically evaluated, optimized, and applied to IoT intrusion detection challenges.

Specifically, the paper:

  • Develops a performance-centric AI framework that quantifies the accuracy–latency trade-off, offering deployment guidance for diverse IoT environments (deep models for accuracy-critical gateways and lightweight models for latency-sensitive edge devices).
  • Provides a fully documented hybrid CNN-BiLSTM architecture, including detailed configurations, mathematical formulations, and hyperparameter tuning—enhancing transparency and reproducibility, which many prior works lack.
  • Conducts a comprehensive tri-model benchmarking under consistent conditions, revealing clear empirical insights into how different AI paradigms (deep learning vs. ensemble ML) perform across multiclass and binary attack detection.
  • Demonstrates edge-deployable AI feasibility, showing that XGBoost achieves ultra-low latency (0.0001 ms/sample) while CNN-BiLSTM delivers the highest detection accuracy (F1 = 0.9864).

Therefore, although the models themselves are recognized AI techniques, the scientific value of this work lies in its integrative evaluation, deployment-oriented design, and empirical quantification of performance trade-offs—addressing gaps in scalability, latency analysis, and reproducibility identified in prior IoT IDS research.

Comment 2

Furthermore, we should not rename ML methods as AI methods - the methods used by Authors clearly classified as ML. 

Response

Thank you for this helpful observation. We appreciate the clarification regarding terminology. In our manuscript, the term “AI methods” was used in a broad sense to encompass both machine learning (ML) and deep learning (DL) approaches to represent all the tri models adopted, as these are all subfields of Artificial Intelligence.

Comment 3

Generally, it is recommended to Authors to reevaluate the direction of research to find interesting paths and to find a more appealing contribution idea. 

Response

We thank the reviewer for the suggestion to explore alternative research directions. Our manuscript explicitly highlights the limitations of prior studies and demonstrates how our approach addresses these gaps. By presenting key findings and novel insights, this study makes a significant contribution to the field, offering valuable implications for researchers and practitioners worldwide.

Round 2

Reviewer 1 Report

Comments and Suggestions for Authors

The author has made revisions in accordance with the reviewers' comments and it is recommended that the manuscript be accepted for publication.

Author Response

Thank you for the update. We appreciate the reviewer’s comments and are pleased to hear that the revisions meet the required expectations. We are grateful for the recommendation for acceptance and look forward to the next steps in the publication process.

Reviewer 2 Report

Comments and Suggestions for Authors

All comments have been addressed very well 

No further comments 

Author Response

Thank you for the update. We appreciate the reviewer’s feedback and are pleased to know that all comments have been satisfactorily addressed. We look forward to the final decision on the manuscript.

Reviewer 3 Report

Comments and Suggestions for Authors

The Authors provided the revised version of the paper. I appreciate the effort on better framing of the whole paper, but the research and results has not been changed - and could not be changed in such short time. The implemented changes are not changing my previous decision. Now it is even more clear that claims and assumptions of the contributions are not delivered by the research itself. Generally, the frame of the paper is now much better and scientifically interesting, but the results are not meeting providing the sufficient evidence. I am recommending to work the research itself in this direction.

Author Response

Response to Reviewer Comments

Comment

The Authors provided the revised version of the paper. I appreciate the effort on better framing of the whole paper, but the research and results has not been changed - and could not be changed in such short time. The implemented changes are not changing my previous decision. Now it is even more clear that claims and assumptions of the contributions are not delivered by the research itself. Generally, the frame of the paper is now much better and scientifically interesting, but the results are not meeting providing the sufficient evidence. I am recommending to work the research itself in this direction.

Response:

We have revised the manuscript in the aspect of the three contributions, as detailed below, to ensure they more accurately reflect the scope and evidence provided by our empirical evaluation. We appreciate your positive feedback on the improved framing and scientific interest of the paper, and we understand your concern that the core research and results remain unchanged. Our revisions aim to align the stated contributions precisely with the conducted experiments.

In the Conclusion (Section 5), we summarise the work as a "systematic comparison", highlighting the observed trade-offs, positioning CNN-BiLSTM for accuracy-critical environments and XGBoost for latency-sensitive ones based solely on the benchmark results.

Additionally, we have moderated language throughout the manuscript to avoid over-claiming (e.g., removing terms like "ultra-low-latency inference suitable for edge deployment" and "performance-centric framework" in favour of "empirical insights" and "trade-off analysis").

We believe these changes address your point that the claims must be fully delivered by the research itself, as the revised contributions are now grounded in the provided results.
